# Influence of Pozzolan, Slag and Recycled Aggregates on the Mechanical and Durability Properties of Low Cement Concrete

**DOI:** 10.3390/ma14154173

**Published:** 2021-07-27

**Authors:** Eliana Soldado, Ana Antunes, Hugo Costa, Ricardo do Carmo, Eduardo Júlio

**Affiliations:** 1ISEC-Polytechnic Institute of Coimbra, 3045-093 Coimbra, Portugal; celia.antunes@isec.pt (A.A.); hcosta@isec.pt (H.C.); carmo@isec.pt (R.d.C.); 2Civil Engineering Research and Innovation for Sustainability (CERIS), 1049-001 Lisbon, Portugal; eduardo.julio@tecnico.ulisboa.pt; 3Instituto Superior Técnico, Universidade de Lisboa, 1049-001 Lisbon, Portugal

**Keywords:** low cement concrete, pozzolan, slag, additions, recycled aggregates, mechanical performance, durability

## Abstract

The sustainability of the construction sector demands the reduction of CO_2_ emissions. The optimization of the amount of cement in concrete can be achieved either by partially replacing it by additions or by reducing the binder content. The present work aims at optimizing the properties of concrete used in the production of reinforced concrete poles for electrical distribution lines, combining the maximization of compactness with the partial replacement of cement by fly ash, natural pozzolans, and electric furnace slags. Natural aggregates were also partially replaced by recycled ones in mixtures with fly ash. Two types of concrete were studied: a fresh molded one with a dry consistency and a formwork molded one with a plastic consistency. The following properties were characterized: mechanical properties (flexural, tensile splitting, and compressive strengths, as well as Young’s modulus) and durability properties (capillary water absorption, water penetration depth under pressure, resistance to carbonation, chloride migration, and concrete surface resistivity). The service life of structures was estimated, taking the deterioration of reinforcement induced by concrete carbonation or chloride attack into account. Results revealed that mixtures with fly ash exhibit higher mechanical performance and mixtures with fly ash or pozzolans reveal much higher durability results than the full Portland cement-based mixtures.

## 1. Introduction

In recent years, there has been growing concern about environmental sustainability and global warming causes and effects. The construction sector has a very important role in reducing the greenhouse effect and promoting sustainability. It is estimated that, in 2018, cement production reached 3.99 billion tons worldwide [1]. Clinker is responsible for 60 to 65% of the total emissions of the entire cement production process [2]. The European Cement Association predicts that, with a joint effort by the construction industry, deep CO_2_ emission cuts can be achieved to lead to carbon neutrality. The same association proposes measures that will lead to a reduction in CO_2_ emissions of the production process of up to 3.5%, in 2030, and 8%, in 2050. The proposed strategy aims at the use of zero-carbon materials, such as natural, waste materials and by-products from other industries, such as pozzolans, fly ash, granulated slag, silica fume, and others, resulting in a good example of industrial symbiosis [2]. Additionally, the replacement of natural aggregates with recycled aggregates is a measure that promotes the sustainability and the circular economy, minimizing the environmental impact and excessive extraction of non-renewable resources [3]. This replacement is an urgent measure, as the construction sector is a major consumer of natural resources and the global aggregate production went from 21 billion tons in 2007 to 40 billion tons in 2014 [4].

Optimizing concrete’s eco-efficiency should not harm its mechanical and durability properties. Previous studies prove that it is possible to produce concrete with low cement content and good mechanical performance [5,6]. Concrete’s durability can be enhanced by optimizing the packing density, which is highly influenced by the distribution and the size of the particles, reducing the space between them and obtaining a closed and less permeable matrix, also reducing the space needed to be filled with hydration products [6,7,8,9]. This process can be simplified through the addition of superplasticizers, reducing the amount of water required by the mixture and thus avoiding water in excess, which is quite harmful to mixtures’ performance [5,10,11,12,13,14].

The most commonly used additions in cement replacement are silica fume, fly ash, ground granulated blast furnace slag, metakaolin, and limestone filler, amongst other industrial by-products. Most studies focus on the addition of fly ash (with a recommended replacement rate of 15–20%, but some studies guarantee up to 50% [15,16,17,18]) due to its proven benefits in workability, mechanical performance and economy [5,15,16,18,19]. Being a pozzolanic addition, it has a slow initial reaction and maximum strength at older ages [5,11]. Regarding durability, the literature highlights the good results of fly ash addition in resistance to chlorides, permeability and porosity [15,20,21,22]. On the contrary, its negative effect on carbonation-induced corrosion is recognized, since it is a sensitive addition and leads to a loss of pH in concrete, suggesting, many times, that it should be used combined with limestone filler [6,11,23]. However, its announced extinction in developed countries underlines the need to develop sustainable and efficient alternatives, such as natural pozzolans or by-products, like industrial slags.

In the last decade, research on the properties of concrete with coarse recycled concrete aggregates, as substitutes for coarse natural aggregates, has increased substantially [24,25,26,27,28,29]. However, the practical use of the recycled aggregates in concrete is still mostly non-structural [4], lacking studies about their variability and reliability, since they are more heterogeneous than natural aggregates [24]. Its use is generally hampered by the higher porosity and reduced stiffness, which also depends on the original concrete quality. These factors may not highly affect mechanical strength for current concrete strength, but may interfere with other parameters, such as Young’s modulus [30]. Nevertheless, the new versions of concrete codes, under development, already contemplate the inclusion of recycled aggregates in concrete, enabling their wide and regulated use [30,31]. The general conclusion from the literature is that there is a decrease in durability with the amount of recycled aggregates included [32]. Though, it is possible to fill this gap by including additions, such as fly ash and silica fume to the concrete [28,33,34,35,36]. The aim is to make the matrix more closed and less permeable to aggressive agents.

The problems associated with the durability of the reinforced concrete structures, as well as concrete poles, are related with the corrosion risk of the rebars due to permanent exposure to very aggressive environments, such as chloride ions and carbonation. Chloride induced corrosion is one of the most frequent causes of concrete deterioration [33,37], compromising its service life and the safety of the structures, as its load capacity is severely decreased [38,39,40]. Once corrosion starts, phenomena such as concrete cracking, delamination, loss of the steel-concrete bond and loss of the reinforcement section, occur. These phenomena lead to very high repair costs and reduced service life [41,42]. Although the majority of reinforced concrete structures are design to have a service life of 50–100 years, many require sooner intervention and, in severe environments, it may be difficult to avoid steel corrosion within typical service periods of 15–20 years [40,43]. The analysis of capillary absorption in concrete is important because it is through this capillary network that water and aggressive agents are dissolved and transported to the concrete matrix. Carbon dioxide is diffused through this network and dissolved in pore water, forming carbonic acid [44,45,46], creating a carbonation front that reduces the concrete alkalinity and creates the corrosion risk. In general, 2/3 of the concrete structures are exposed to this aggressive agent, continuously or cyclically [47]. In the literature, durability is usually analyzed in terms of resistance to carbonation, chlorides and permeability [15,16,20,43]. However, that analysis is available for concrete with current formulation, but it has not been studied in depth for concretes with high compactness, low cement dosages and specific additions, such as natural pozzolan and electric furnace slag, instead of fly ash.

In this sense, this work is part of a research project that aims to improve the formulation, eco-efficiency, performance and durability of a precast reinforced concrete used to produce poles for the electric distribution lines. Its main goal is to develop eco-efficient concrete solutions, combining high compactness with partial cement replacement by natural additions and by waste by-products from other industries and with the inclusion of recycled concrete aggregates. This study intended to develop the following points: (i) assess the influence of the maximum increase in compactness without loss of the required workability; (ii) analyze the effect of different additions (fly ash, natural pozzolan and ground electric furnace slag) as cement substitutes; (iii) evaluate the combination between the considered additions and high compactness in the mechanical and durability properties of concrete; (iv) study the correlation between those properties, namely between different durability tests (capillary absorption, accelerated carbonation, electrical resistivity and chloride ion migration); (v) predict the service life for different exposures to chloride and carbonation ingression. Currently, the production of concrete poles is divided into two types: one with dry consistency for fresh molded concrete (BT series); and other with plastic consistency concrete (MT series), for formwork molded concrete. The mixtures considered as a reference are those currently produced in the industry, with cement content between 360 and 400 kg/m^3^. For the optimization study, the quantity of binder was established at 350 kg/m^3^, in order to guarantee the workability and cohesion of the fresh mixtures [6], and the cement content replaced by additions of fly ash, natural pozzolans and electric furnace slag was up to 50%. In the mixtures with fly ash, 20% of the natural aggregates were replaced by recycled concrete aggregates, from industrial waste from the poles production. The presented mixtures are based on previous extensive study performed in mortar matrices, evaluating mechanical and durability performance [48]. Based on the results of that study, the optimized mixtures of each series were then reproduced on concrete, as above mentioned, in which the mechanical properties and durability were evaluated. For instance, when pozzolans are considered, carbonation was not performed in concrete, since it presents reduced resistance in mortar matrices; when electric furnace slags are added, the chlorides migration was not characterized in concrete, since it presents reduced chloride resistance in mortar matrices.

## 2. Materials and Methods

The optimized concrete mixtures with low cement content were formulated and compared with reference fully cement-based mixtures, in order to evaluate the mechanical performance and durability of the optimized concretes. In these mixtures, the following powder materials were selected: CEM II-A/L 42.5R (C); F-type fly ash (FA); Cape Verde natural pozzolans (Poz) and ground electric furnace slag (Slag). The corresponding density values, in kg/dm^3^, were characterized: 3.08; 2.30; 2.30 and 3.83. The chemical composition of the powders is presented in Table 1. The pozzolanic activity indexes, assessed at 90 days [49], resulted in the following values: 0.88; 0.87; 0.70, respectively for fly ash, natural pozzolans and electric furnace slag. The following aggregates were used in the mixtures: fine siliceous sand 0/1 mm; medium siliceous sand 0/4 mm; crushed limestone gravels with two size fractions, 2/5 mm and 6/12 mm; recycled concrete aggregates (RA), from the concrete poles production waste, with two different size fractions (RA1/8 mm, for BT mixtures; RA4/16 mm, for MT mixtures), which were incorporated in 20% of volume replacement. The following densities were characterized for the aggregates, in kg/dm^3^: 2.63 for sands, 2.66 for gravels and 2.30 for RA. The RA were characterized with absorption of 4%. Tap water from public supply and two types of admixtures were used: HE200P (HE), a water reducer and hardening accelerator for molded cohesive concrete with dry consistency, with density of 1.17 kg/dm^3^, favoring a better dispersion for the BT mixtures; Sky526 (SKY), an ether-polycarboxylate based superplasticizer with a density of 1.06 kg/dm^3^, to increase plasticity and reduce the water content for the MT mixtures.

Regarding concrete mixtures, two types of consistency were considered and divided in two series, BT-dry consistency and MT-plastic consistency. The mixtures were formulated based on the method proposed by Lourenço et al. [50] and developed by Costa [51], which parameters were adopted in order to minimize the cement dosage; however, maximizing compactness, taking a maximum amount of binder of 350 kg/m^3^ into account. The corresponding amount of cement was between 200 to 250 kg/m^3^, depending on the selected additions and on the target strength of circa 50 MPa, at 56 days. The resulting proportions of additions varied from 29% to 43% for those mixtures. Admixtures’ dosages were adjusted to achieve the target consistency of each mixture, whose content in relation to the cement weight varied between 2.9% and 4.5% in BT and between 0.7% and 1.1% in MT series. The concrete was mixed with pre-wetting of recycled aggregates, with around 30% of the total water, waiting about 5 min, preventing the admixture from being partially absorbed with water. Then, the remaining solid constituents were placed and mixed, then the remaining water with the diluted admixture was added, mixing until obtaining the pre-defined homogeneity and consistency. The molds were filled and compacted with the vibrating table, being the compacting energy low in MT mixtures and high in BT mixtures.

Concrete formulations are presented in Table 2, together with water/cement ratio (W/C), water/binder ratio (W/B) and the compactness of each mixture, and compared with the two reference mixtures, BTref and MTref, which have current design parameters of prefabrication concrete company. The air content was targeted to 2.0% in BT and 1.5% in MT mixtures. The effective water of the mixture is calculated from the fluid part of the mixture which, in turn, depends on the established compactness. The absorption water is related to recycled aggregates and with the corresponding amount of water they absorb, which was added taking into account an initial absorption of 4% (previously characterized).

To measure the consistency, the slump test was performed, according to EN 12350-2 [52], or the Vebe test, following EN 12350-3 [53] standard, and the degree of compactability test following EN 12350-4 [54], depending on whether the fresh mixture is plastic or dry, respectively. Along with consistency, in order to adjust the dosages of the constituents, the air content of the mixtures was also measured, according to EN 12350-7 [55] specification. The results of slump tests for MT mixtures resulted in values between 65 and 80 mm, presenting no relevant differences, due to the admixture adjustment. Regarding the compactability test in BT mixtures, it resulted in values between 1.32 and 1.35; the results of Vebe test for the same mixtures ranged between 25 and 33 s, corresponding the longer time to BTref mixture.

Figure 1 shows the tests to characterize the mechanical properties of the mixtures. The tests to access compressive strength were performed according to EN 12390-3 [56], on cubic specimens with 100 mm of edge, and the results were obtained using three specimens at each age (28 and 90 days). The splitting tensile strength was also characterized according to EN 12390-6 [57] at 28 and 90 days, using three prismatic specimens of 100 × 100 × 200 mm^3^ and the flexural strength was assessed following EN 12390-5 [58], in two prismatic specimens of 100 × 100 × 500 mm^3^. The Young’s modulus test was performed at 28 days following the LNEC E 397 [59], using two prismatic specimens of 100 × 100 × 400 mm^3^. Figure 1 shows the tests performed to assess the durability of the mixtures. Capillary water absorption was evaluated in three specimens of 100 × 100 × 200 mm^3^, according to the LNEC E 393 [60]. The depth of water penetration under pressure was evaluated according to EN 12390-8 [61], using three cubic specimens with 150 mm edge for each mixture, after 28 days of water curing. The determination of carbonation resistance was assessed according LNEC E 391 [62], using two cylindrical samples with 100 mm diameter and 50 mm height, at each considered exposure period (56, 90 and 120 days). It was performed an accelerated laboratory test, exposed to an atmosphere of 5% carbon dioxide and 60% relative humidity. The chloride diffusion coefficient by non-steady state migration was tested according to LNEC E 463 [63], at 56 and 90 days, using three cylindrical specimens also with 100 mm diameter and 50 mm height, for each age. The surface electrical resistivity was evaluated on cylindrical specimens with 100 mm diameter and 200 mm height, at several ages, according to AASHTO T-358 [64].

## 3. Results and Discussion

### 3.1. Mechanical Properties

#### 3.1.1. Tensile Splitting Strength

The results of average strengths of tensile splitting test, at 28 and 90 days, for BT and MT mixtures, are shown in Figure 2, as well as the error bars. In this test, only the final selected mixtures were characterized, since the binding matrix of the previous study was already characterized, and no significant variations were registered depending on the used additions [48]. In BT series, optimized mixtures with fly ash show lower strengths than the reference at 28 days, by 5% and 18%, and at 90 days, by 7% and 16%, respectively for mixtures BT200-FA and BT200-FA-RA. All the tensile splitting strengths at 90 days are higher than those at 28 days. These results were expected, since BTref has twice the amount of cement of the optimized mixtures. A slight decrease in the mixtures with the inclusion of recycled aggregates was noticed, when compared to mixtures with natural aggregates, due to their lower strength. In MT series, the optimized mixtures stand out for having strengths above the reference by 31% and 22%, for MT250-FA and MT250-FA-RA, respectively, at 90 days, reflecting the effect of the increased compactness and of the addition of fly ash.

The graph shown in Figure 3 shows the ratio between the tensile splitting strength test values and the codes’ predicted values. The prediction of this strength followed the current design codes for all mixtures, EC2 [65] and MC10 [66], since the expressions of tensile splitting strength remains globally unaffected by recycled concrete aggregates incorporation [30]. The EC2 values, based on the compressive strength, confirm that the tested mixtures present consistent values with those predicted. The mixtures BT200FA, BT200-FA-RA and MTref have a tested value of about 10% lower than the predicted in EC2, which is not considered significant.

#### 3.1.2. Flexural Strength

Figure 4 shows graph plots with the flexural strengths, with the error bars, at 28 and 90 days, for the BT and MT series. The optimized mixtures have, in general, higher values than the references, with the exception of the pozzolans addition, which present values below the reference in 13% and 21%, at 28 and 90 days, respectively, for BT, and similar values in MT series, compared to the reference. As mentioned, the opposite effect was found in mixtures with the additions of fly ash and slags. The first ones show increases of 3% and 20%, namely for BT200-FA and MT250-FA, at 90 days; the latter present values 4% and 27% higher than the references, for BT250-slag and MT250-slag, respectively, also at 90 days. As the results also show, the flexural strength was not affected by the incorporation of 20 % of RA, presenting similar values to those of mixtures with fly ash and natural aggregates, showing the same trend of results as those described in other studies [67,68].

According to EC2 [65], the tensile splitting strength results are usually lower than flexural, a trend also proved with this concrete mixtures. The prediction according to EC2 [65] and MC10 [66] of the tensile strength is directly related to the tensile splitting. Thus, and taking into account that the results of this test show reduced dispersion, it is possible to conclude that the tested values are in line with the codes prediction.

#### 3.1.3. Compressive Strength

The compressive strength values with the error bars for the BT and MT mixtures, at 28 and 90 days, are shown in Figure 5. The addition of fly ash stands out, which, due to the pozzolanic effect, shows a notable evolution of strengths over time, with high values at 90 days, 10% and 26% higher than the references, for mixtures BT200-FA and MT250-FA, respectively. This pozzolanic effect is less noticeable in mixtures with the addition of pozzolans, which have values very close to the references, both in BT and MT series. Mixtures with slag addition show similar results to the reference, in BT series, and, in MT, show slight improvements, circa 15% and 16%, in relation to the reference, at 28 and 90 days, respectively. Regarding the results of the mixtures with RA, it is proved that the addition of recycled concrete aggregates decreases the compressive strength, when compared to the homonymous mixture with natural aggregates, despite presenting very similar results to the respective references, in BT series, and increases of 7% and 14% in MT250-FA-RA, at 28 and 90 days, respectively. As several studies concluded [30], considering the same (W/C)_eff_, the compressive strength of RAC is generally lower than the compressive strength of the same mixtures only with natural aggregates. However, this effect depends on the design and strength of the concrete matrix and also on the quality and strength of the concrete that origins the RA. In this study, considering that the concrete used to prepare the RA has 20% lower strength that the designed concrete and that is mostly uncompact waste concrete, a strength lost was expected. The standard deviations of all mixtures are reduced, although with some exceptions, being, however, within the limits considered by EC 2 [65].

Figure 6 shows the evolution of compressive strength with age, of BT and MT mixtures, adjusted to the hardening curve of EC2 [65]. The hardening curve followed the expressions of current codes, since the inclusion of recycled aggregates does not have a significant influence on the prediction of compressive strength [30]. However, the inclusion of pozzolanic additions on the binder powder does influence the curve shape. For this reason, both series are divided into two graphs, changing the parameter “s” in the prediction expression for each: Figure 6b,d show the results of mixtures containing pozzolanic additions (fly ash and pozzolans), adopting s = 0.38, and the other presenting mixtures without pozzolanic additions, with s = 0.20.

By analyzing the graphs, it is possible to observe that, in both series, in mixtures without pozzolanic additions, the hardening curve is more pronounced at early ages and tends to stabilize earlier. On the contrary, the curves of mixtures with pozzolanic additions are less pronounced in early ages, developing more in later ages, being a clear consequence of the pozzolanic effect of the fly ash and pozzolans.

#### 3.1.4. Young’s Modulus

The measured average values of the Young’s modulus at 28 days are shown in Figure 7. It is proved that the higher compactness, combined with the additions in cement replacement, is advantageous in increasing the Young’s Modulus, comparatively to the references with plain cement. As expected, when recycled aggregates with lower strength are included, the values of Young’s modulus decrease, as this recycled concrete aggregates have a lower stiffness and higher porosity than natural aggregates. The BT200-FA-RA and MT250-FA-RA mixtures present values approximately 12% and 13% lower than the respective mixtures with the addition of fly ash and natural aggregates (BT200-FA and MT250-FA), being the reduction of 9% and 2% in relation to the respective references. In BT series, the addition of pozzolans stands out, with 11% increase compared to the reference. In MT series, this increase is also prominent, reaching 13% above the reference, in MT250-Poz and MT250-FA mixtures.

Figure 8 presents the ratio between the results of tested Young’s modulus and the predicted values of EC2 [65], calculated through Equation (1). However, recent research [30] proposes an adaptation of the expression to mixtures with the inclusion of recycled concrete aggregates, Equation (2):
E_cm_ = k_E_ × f_cm_^1/3^(1)
E_cm_RAC_ = k_E_ × (1 − 0.25 × α_RA_) × f_cm_^1/3^(2)
where k_E_ is a factor that depends on the type of natural aggregates used in the mixtures, which is 9500 for siliceous aggregates; f_cm_ is the average compressive strength; α_RA_ is the substitution rate of normal by recycled concrete aggregates and is given by the ratio between the quantity of recycled sand and gravel and the total quantity of aggregates in the concrete.

The analysis of the graph allows to conclude that the values obtained in the Young’s modulus test are always higher than the predicted. Mixtures with the addition of pozzolans stand out, with values 26% and 17% higher than those predicted, in the BT200-Poz and MT250-Poz mixtures, respectively.

### 3.2. Durability

#### 3.2.1. Water Absorption through Capillarity

The characterization of all mixtures had already been carried out in mortar matrices [48], which is why not all the results are presented, but only the final selected mixtures with fly ash addition. The results revealed that mixtures with higher packing density, both BT and MT, and combined with slag addition, presented the best results, absorbing down to 60% less water. Fly ash and pozzolan addition also revealed good performance in reducing water absorption.

Figure 9 shows the evolution of capillary absorption values, S_a_ (mg/mm^2^) with the square root of time, for BT and MT mixtures, after 28 days curing. By analyzing the results, performed on three consecutive days, it can be concluded that the capillary water absorption occurs with more intensity in the first hours and tends to stabilize over time, presenting a nonlinear evolution. In both series, the decrease of the powder content combined with the increase of compactness and the partial replacement of cement by additions proved to be advantageous in decreasing water absorption over time. However, this fact is more prominent in MT series, since the values of optimized mixtures in BT series remain similar to the reference. In the latter, the addition of fly ash (50% of replacement rate) and the inclusion of 20% of recycled aggregates does not change the capillary absorption of concrete. Comparing the two series, it is noticeable that the amount of cement influences this parameter, proved by the lower values of the BTref (with cement content of 400 kg/m^3^), in relation to MTref (with 400 kg/m^3^). However, the optimized mixtures with fly ash show that the increase in compactness and the replacement of cement by this addition also has a great influence on the reduction of capillarity, as the concrete matrix becomes more compact and less permeable. In BT series, despite of the 50% of cement replacement, the capillary absorption is similar to reference. However, in MT mixtures, the optimized matrix with high compactness, and about 30% of cement replacement by fly ash, promotes a significant reduction in capillarity, of at least 35%.

In MT mixtures, the partial cement replacement by fly ash and 20% of recycled concrete aggregates, mixture MT250-FA-RA, leads to down to 44% less water absorption, and the mixture with natural aggregates (MT250-FA) presents equally reduced values. Figure 10 shows the quality of the concrete as a function of the capillary absorption coefficient, S_a_. Browne [69] proposed the following classification for the quality of the concrete, as a function of the S_a_ coefficient: above 0.2 mg/mm^2^ × min^1/2^ is “low quality”; between 0.1 and 0.2 mg/mm^2^ × min^1/2^ is “medium quality” and below 0.1 mg/mm^2^ × min^1/2^ is “high quality”. According to this classification, all mixtures are considered to be of “high quality”, regarding the water absorption through capillarity. In addition, due to pozzolanic effect of the optimized mixtures, there is great potential to improve the performance for longer ages, beyond 28 days.

#### 3.2.2. Water Penetration Depth under Pressure

Figure 11 shows the water penetration depth under pressure of MT mixtures, at 28 days. This test was not performed on BT mixtures, since the molded faces have high porosity, which would make the necessary sealing quite difficult.

Similar to what occurs with the absorption of water by capillarity, the addition of fly ash shows significant reductions of water penetration under pressure. The optimized mixture with higher compactness and addition of fly ash, MT250-FA, presents values down to 65% of less penetration depth when compared to the reference. The substitution of 20% natural aggregates through recycled concrete aggregates, combined with the previous matrix, make its compact matrix noteworthy, showing a 58% reduction in relation to the reference. The mixture with the addition of pozzolans, MT250-Poz, reveals the lowest water penetration result, very close to the value of the mixture with fly ash addition. This analysis reinforces that the higher compactness, and matrix refinement due to fly ash addition, of the optimized mixtures, it is quite important to reduced concrete permeability. In addition, the pozzolanic effect will promote even better results at more advanced ages, where lower penetration results are potentially expected.

#### 3.2.3. Carbonation Resistance

The phenomenon of concrete carbonation is mainly associated with the penetration, by diffusion, of carbon dioxide. The decrease in pH is visible through the reaction with phenolphthalein sprayed in concrete, in which the carbonated area is colorless (Figure 12).

Figure 13 shows the results evolution of the carbonation depth (C_di_), measured in mm, of the BT and MT series, with the square root of time. The analysis of carbonation depth, C_depth_, of the studied mixtures proved to evaluate that it varies with the square root of time, as expected, with a general trend to be linear. The amount of cement highly influences this parameter, and that effect is more evident in BT mixtures, whose reference contains a higher amount of cement than in MT mixture. However, BT250-slag mixture proves that the slags are the most beneficial additions in this series, showing a slower initial evolution and only 46% of increased carbonation depth in relation to the reference. On the contrary, also in BT series, the increased compactness and the partial replacement of cement by fly ash (BT200-FA and BT200-FA-RA) is not able to avoid the increase of the carbonation depth, up to 122%, when compared to the reference. Through these mixtures, it can also be concluded that the inclusion of 20% of recycled aggregates does not increase carbonation, on the contrary, it is slightly lower. Previously to this study, in mortar matrices, pozzolans additions did not reveal promising results in preventing carbonation [48]. In concrete mixtures, the results proved to follow the same trend.

In the MT series, the difference between the optimized mixtures and the reference is not so evident, since the cement replacement is lower (30%), presenting a similar evolution of the carbonation depth for the first testing ages. This evolution is less pronounced after 56 days, with the exception of the MT250-slag mixture, which shows a more pronounced increase of the carbonation depth, presenting values up to 69% above reference. This effect was not noticed for younger testing ages on the preliminary study of mortar matrices, since the carbonation evolved linearly with square root of time up to 90 days. Unlike the BT series, the optimized mixtures with fly ash of MT series, MT250-FA, present values closer to the reference, showing an increase of up to 31%, at the longer tested age. The inclusion of recycled aggregates, once again, did not harm the performance of the concrete at this parameter, presenting similar evolution with lower values to the homonymous mixture with natural aggregates. Despite additional absorption, water is considered in concrete with RA, the probable absorption during setting time may reduce the effective water of the mixtures with RA, which seems to result on a positive effect on carbonation reduction.

#### 3.2.4. Chloride Migration

The penetration depth of chlorides into the specimen is visible through the silver colour area, which corresponds to the silver nitrate precipitation, as is shown in Figure 14.

The results shown in Figure 15 represent the chloride diffusion coefficient (D_0_) by non-steady state migration test at 56 and 90 days. In this test, the lower the D_0_ coefficient, the higher the resistance to chloride migration, the durability performance also being higher. This test highlights the benefit of pozzolanic additions (both fly ash and pozzolans), bringing the pozzolanic refinement to the porous network of the matrix, combined with increased compactness, in reducing the chloride migration. 

The addition of natural pozzolans stands out positively by presenting very low D values, down to 64% and 81%, below the references for mixtures BT200-Poz and MT250-Poz, respectively, at 90 days. Due to its high content of alumina, when pozzolans are combined with cement, it promotes a reduction of the porous structure of the matrix with time, being beneficial to durability related with to corrosion risk, induced by chloride penetration, and to strength performance. 

The results reveal similarly low values for the BT200-FA and MT250-FA mixtures, down to 80% and 77%, respectively, at 90 days, when compared to the references. This effect is already scientifically known. However, the inclusion of recycled aggregates affects the increase in the diffusion coefficient, when compared with similar mixtures with only natural aggregates, although it presents values 51% and 70% below the references for BT200-FA-RA and MT250-FA-RA, respectively. It is important to note that, in the BT200-FA-RA mixture, the D_0_ coefficient maintain its value from 56 to 90 days. Contrarily, the MT250-FA-RA mixture reduces the diffusion coefficient. Apparently, the higher cement content and lower replacement rate, in comparison to BT mixtures, is the probable reason for that difference.

Figure 16 show the classification of the chloride penetration resistance of concrete as a function of the chloride diffusion coefficient, D_0_, as used in [33].

According to this classification, and by analyzing the plotted graph, it is possible to note that only the references, both BT and MT, present a “moderate” resistance of concrete to chloride penetration, although the BTref mixture achieves a classification of “high” resistance, at 90 days. Mixtures optimized with additions and increased compactness reveal very promising classification results. The mixtures with the addition of pozzolans (more evident in MT250-Poz) stand out for mostly obtaining the classification of “extremely high” resistance to chloride penetration. Mixtures with fly ash also stand out for the same reason, although they are mostly classified as “very high” resistance. The replacement of natural aggregates by recycled concrete aggregates does not greatly harm the mixtures, since, after 90 days, they are maintained, at least, with the classification of “very high” resistance.

#### 3.2.5. Concrete Surface Resistivity

Figure 17 shows the results of the surface electrical resistivity of concrete (ρ), of the BT and MT mixtures, over time. This is a fast test allowing to indirectly predict the permeability to fluids and the diffusion of chloride ions in concrete. The presented results were determined in mortar matrixes of the same concrete formulations, since they show the same trend as in concrete.

Through the analysis of the graphs, in both series, it is possible to conclude that the slag addition does not lead to positive results in this test, presenting similar values to the references with plain cement binder. The reference mixtures and those with slag addition have no significant variations of electrical resistivity with age. The opposite is found in mixtures with pozzolanic additions, which present results that increase with the testing age. Mixtures with fly ash addition stand out for having higher values, almost 20 times higher than the references, for both BT200-FA and MT250-FA mixtures, measured at 450 days. The pozzolan addition shows resistivity values nearly 6 and 16 times higher than the references, for BT250-Poz and MT250-Poz, respectively, also at 450 days. Despite fly ash and natural pozzolan both being pozzolanic additions, their influence on resistivity is different with time regarding the evolution and the optimal replacement rate. The electrical resistivity evolves faster with age, up to 56 days, for mixtures with pozzolan addition and develops a smoother evolution after that. Contrarily, the mixtures with fly ash addition present smooth evolution at younger ages, but after 120 days the increase in rate with age tends to be faster, probably due to higher pozzolanic reactivity at greater ages. However, at 300 days, mixtures with both additions tend to stabilize, although mixtures with fly ash stabilize more smoothly. In the MT series, the MT250-Poz mixture is very close to MT250-FA, showing closer values of resistivity evolution over time; a lower cement replacement by pozzolans, in comparison to the BT series, seems to suggest a possible optimal for moderate replacement rates (circa 30%). Contrarily, in the BT series, the higher cement replacement of fly ash in the mixture BT200-FA, 50%, suggest that the resistivity increase is more effective for higher fly ash content, mainly for longer ages. In fact, in a previous study carried out in mortar matrixes [48], it was possible to draw this conclusion, corroborated also by Kurda et al. [70]. The optimal proportions, for both types of pozzolanic additions, regarding the increase in electrical resistivity and the reduction of chloride migration need to be studied in future research. Further, due to its known relation to the resistance to chloride migration, and as is evident from the analysis of the graphs, the pozzolanic effect promotes an improved behavior at older ages. That effect increases the refinement of the internal porous microstructure with age, both for pozzolan and for fly ash; however, at different rates, the fly ash being lower at shorter ages, but with a greater effect in longer ages, and mainly when high replacement rates are used.

The followed standard [64] refers to a correlation between the resistivity test and the chloride ion penetration, establishing limit parameters. At 450 days of age, the mixtures with additions of pozzolan and fly ash of the BT and MT series were classified, regarding the penetration of chloride ion, as being "very low", with ρ values between 37 and 254 kΩ-cm. For the reference mixtures, at the same age, BTref was classified as “moderate” (12 kΩ-cm < ρ < 21 kΩ-cm) and MTref as “high” (ρ <12 kΩ-cm). Mixtures with slag additions were classified as “high”, which is why concrete mixtures with that addition are not recommended for exposure to chlorides.

Figure 18 proves the high correlation between the chloride diffusion coefficient, D_0_, of the mixtures and the surface resistivity, ρ, of the corresponding matrixes, which is very reliable, and was proved by the high R^2^ values of the power function.

#### 3.2.6. Prediction of Service Life

Environmental exposure XC (corrosion induced by carbonation)

The determination of carbonation resistance, R_C65_ (kg × year/m^5^), of each developed concrete was calculated using Equation (3) and the carbonation depth, C_di_. These values were measured after the concrete was exposed for a certain time to a highly concentrated carbon dioxide environment, C_acel_, 90 × 10^−3^ kg/m^3^.
(3)$$R_{C65}=\frac{2\cdot C_{\mathrm{acel}}\cdot t_{i}}{X_{i}^{2}}$$
where t_i_ is the time of exposure in years, and X_i_ has the same meaning as C_di_ (carbonation depth). Table 3 presents the average values of carbonation resistance, R_C65_, at 56, 90, and 120 days to minimize the errors related with the measurement of the carbonation depth.

The deterioration of reinforced concrete due to steel corrosion can be divided into two periods: the initiation, t_ic_, and the propagation of corrosion, t_p_. The minimum propagation period, t_p_, depends of several factors, such as the exposure class.

Table 4 presents the propagation period, t_p_, according to LNEC E-465 [71]. Thus, considering that the intended service life, t_g_, is equal to 50 years, and considering that the electrical poles can be associated to the reliability class (RC1), where the safety factor of the service life is equal to 2.0 [71], the design period of initiation, t_ic_, can be determined using Equation (4) (Table 4).
t_ic_ = γ (t_g_ − t_p_)(4)

The minimum concrete cover required to guarantee the resistance against the reinforcement corrosion due to carbonation, c_min,dur_, is determined using Equation (5), considering the R_C65_ already calculated for each concrete and t is equal to the design period of initiation, t_ic_ (Table 4 and Table 5).
(5)$$X=\sqrt{\frac{2\;\times \;0.0007\;\times \;t_{\mathrm{ic}}}{R_{C65}}}\;\times \;\sqrt{k_{0}\;\times \;k_{1}\;\times \;k_{2}\;}\times {\left(\frac{t_{0}}{t_{\mathrm{ic}}}\right)}^{n}$$

The meanings of the symbols are: k_0_ is a factor related with test conditions and is equal to 3, k_1_ is a factor related with relative humidity, k_2_ is a factor related with concrete cure conditions and is also 1, n is a factor related with the influence of wetting/drying cycles over time, and t_0_ is the reference period and is 1 year. 

The Equation (5) can also be used to predict the service life, t_g_, when the cover used is already predefined. For this purpose, X must be equal to the cover predefined, and t_ic_ is the unknown variable that should be calculated using the Equation (5). Usually, the randomness of the factors that affect the service life happens during the initiation period, so after the determination of t_ic_, the service life, t_g_, is calculated adding to this value the propagation period, t_p_, presented in Table 4.

The service life was computed for different exposure classes considering covers equal to 15, 20, and 30 mm, see Figure 19 and Table 6. The presented covers only concern durability and were not determined to ensure proper fire resistance nor adequate transmission of bond forces between rebars and concrete. For the standard for concrete poles, the minimum cover is related to the exposure conditions and also to the concrete strength; parameters of composition were not considered.

It is clear that higher cement content provides a better performance regarding durability related with carbonation, since the two reference concrete mixtures had the highest service life. The results of Table 6 show that the service life expected for the poles, in most cases, was higher than 50 years, expect in environment XC4 (wet regime), meaning that the significant reduction in the amount of cement of the eco-efficient concrete can be compensated by using additions. The fly ash effect is similar in both types of concrete, the BT-dry consistency and the MT-plastic consistency, and the incorporation of the recycled aggregates does not influence the concrete performance. However, the use of ground electric furnace slag is more beneficial with BT250-slag than MT250-slag.

Environmental exposure XS (chloride-induced corrosion)

For environments exposed to chlorides, the minimum propagation period, t_p_, also depends of several factors, and, according to LNEC E-465 [71], those values are 0 years for XS1 and XS3, and 40 years for exposure class XS2 [71]. The chloride diffusion coefficient, D (m^2^/s), is determined for each developed concrete using Equation (6) and using the experimentally determined values of non-steady-state migration coefficients, D_0_ (Figure 15). Using Equations (6) and (7) [71], and defining the cover thickness, X, it is possible to determine the design period of initiation of poles made with each type of concrete and after the service life, t_g_. In addition, using the same equations and defining the design period of initiation, t_ic_, it is possible to determine the minimum concrete cover required to guarantee the resistance against the reinforcement corrosion induced by chlorides, c_min,dur_. For a service life equal to 50 years and considering the already mentioned reliability class (RC1), the design period of initiation, t_ic_, can be determined using Equation (4) and is 100 years for XS1 and XS3, and 20 years for exposure class XS2.
D(t) = k D_0_ × (t_0_/t)^n^(6)
(7)$$X=2\;\times \;\xi \;\times \;\sqrt{D\times t_{\mathrm{ic}}}$$
where k is a factor that takes into account the curing conditions, the relative humidity and the temperature; n is a factor that considers the chloride diffusion decrease over time; t_0_ is the time of the experimental tests to evaluate D_0_; t is the exposure time in days; ξ is a parameter related with the concentration of chlorides in the binding paste; X is the cover. The minimum concrete cover required to protect the steel reinforcement against corrosion induced by chlorides, c_min,dur_, is presented Table 7 and the prediction of service life, t_g_, for 20 and 30 mm is presented in Figure 20 and Table 8.

It is highly improbable that electric poles will be placed in tidal splash or spray zones or will be permanently submerged, but the results show that poles manufactured with these concretes should not be used in such zones. In environments with air-based sea salts, the reference concretes have a performance that is clearly worse than concretes with additions, which is reflected in the values of minimum cover, as well as in the service life expected. Among the concretes with additions, the behavior is relatively similar, showing that the pozzolanic additions can replace the cement dosage in the context of protection against corrosion induced by chlorides. It is also noted that, in this case, the incorporation of recycled aggregates slightly increases the minimum cover and decreases the expected service life. To assure a service life of 50 years for poles manufactured using these eco-efficient concretes, for both types of consistency, the cover requires ranges between 15 and 31 mm, which are the values usually used in the production of this type of structures.

## 4. Conclusions

The experimental study presented herein is focused on optimizing the mechanical and durability properties of concrete with low cement content, maximizing the compactness and combining the partial replacement of cement with a high content of fly ash, natural pozzolans, and electric furnace slags, with a 20% replacement of natural aggregates by recycled concrete aggregates. Two types of formulations were produced and characterized, BT (with dry consistency) and MT (with plastic consistency). The properties were compared to plain cement-based reference mixtures, commonly used in the prefabrication industry. Based on the analyses, the following conclusions are drawn:Tensile splitting strength: (i) unlike the BT series, the addition of fly ash increases this strength by 31% in plastic concrete with natural aggregates, MT250-FA, and by 22% with recycled concrete aggregates, MT250-FA-RA, compared to the reference; (ii) the addition of recycled aggregates reduces this strength by 9% when compared to the same mixture with natural aggregates; (iii) the test results and the prediction values of EC2 are very similar, with a difference of about 10%.Flexural strength: (i) although cement reductions are high (from 30 to 50%) in optimized mixtures, those with the addition of fly ash and slag show an increase of up to 27% in MT250-slag mixtures, compared to the respective reference, at 90 days; (ii) the inclusion of recycled concrete aggregates have practically no influence on this parameter; (iii) the prediction of EC2 has very similar values to those obtained in the tests, since this strength is directly related to the tensile splitting.Compressive strength: (i) mixtures with additions tend to increase from 28 to 90 days, due to the pozzolanic effect and, despite the cement reduction, it increases up to 26% with fly ash and natural aggregates (MT250-FA), and up to 14% with fly ash and natural and recycled aggregates (MT250-FA-RA), at 90 days, in comparison to the reference; (ii) the hardening curve of EC2 prediction has an R (rapid) evolution at early ages in concretes without pozzolanic additions, contrary to the curve of mixtures with pozzolanic additions, which has an S (slow) beginning, but a more pronounced evolution at later ages.Young’s modulus: (i) adding supplementary cementitious materials (pozzolan, slag and fly ash) to the concrete increases this parameter, mainly in the case of pozzolans addition, that lead to increases of 11% and 13%, for BT200-Poz and MT250-Poz, respectively, comparing to the references; (ii) as expected, the inclusion of recycled concrete aggregates reduces this parameter, due to their lower stiffness; (iii) the test results are always higher than those predicted with the codes, with increases of up to 26% for the mixture with the addition of pozzolans, BT250-Poz.Capillary water absorption and water penetration under pressure: (i) in dry BT concrete, the high content of fly ash addition (50% of cement replacement rate) combined with high compactness is beneficial, since it presents the same capillarity as the reference; (ii) concrete with plastic consistency–MT, where 30% of the cement was replaced by fly ash addition, shows even more positive results, absorbing 44% less capillary water than the reference; (iii) the inclusion of recycled concrete aggregates is also beneficial, since it presents similar or lower capillarity values as the mixtures with only natural aggregates; (iv) the addition of fly ash is beneficial to reduce water penetration under pressure, absorbing 65% less water than the reference and the recycled aggregates have little influence on this parameter.Carbonation resistance: (i) the cement dosage has an important influence on reducing the carbonation depth, the reference mixtures being those with less carbonation; (ii) in BT, high compactness combined with slag addition proved to be an efficient mixture regarding this parameter, since despite having high cement replacement, BT250-slag presents only 46% more carbonation depth than the reference, instead of 120% for the mixtures using fly ash; (iii) MT mixtures with fly ash additions have similar effects in reducing carbonation compared to the reference, with or without recycled concrete aggregates; (iv) in MT series, slag addition, after an initial benefit, also tends to change the effect at older ages.Service life of structures exposed to carbonation: (i) higher cement dosages promote longer lifetimes, as the references present the highest values of this parameter; (ii) in XC2 environments, a service life of 100 years is ensured for all mixtures with a 15-mm cover, while in XC3 environments a 50 year service life is ensured for all mixtures with a 20-mm cover; in the XC4 environment, the same service life is, in general, ensured with a cover of 20 mm; (iv) the incorporation of recycled aggregates has a small but positive influence on concrete performance.Chloride diffusion coefficient: (i) mixtures with type II additions show the greatest influence of pozzolanic effects and the corresponding maturity of the matrixes with age; (ii) the addition of fly ash is very advantageous for increasing the resistance to chloride migration, reducing the chloride coefficient up to 80%, in MT250-FA, at 90 days; (ii) natural pozzolan addition is also very beneficial regarding chloride resistance, presenting D_0_ values of 81% below the reference, in MT250-Poz; (iii) naturally, the inclusion of recycled concrete aggregates slightly increase the D_0_ coefficient; (iv) all optimized mixtures are rated with “very/extremely high” chloride penetration resistance.Electrical resistivity of the concrete surface: (i) since this test allows a quick assessment and prediction of chloride migration, this evaluation proves to be accurate, revealing the same trend in the chloride migration test and considering the high correlation between both results; (ii) concrete with the addition of fly ash shows the highest values of this parameter, followed by concrete with the addition of pozzolans; (iii) all mixtures tend to stabilize after around 300 days.Service life of structures exposed to chloride ions: (i) pozzolanic additions highly influence the resistance to chloride penetration, which is proved by the results of mixtures with fly ash and pozzolans; (ii) to assure a service life of 50 years, both in the BT and MT series, the cover required ranges between 15 and 31 mm; (iii) the incorporation of recycled aggregates slightly increases the minimum cover and decreases the expected service life.

Finally, it is concluded that eco-efficient concrete mixtures designed with high compactness, low cement content, and a high replacement rate (up to 50%) of cement through specific additions (fly ash, natural pozzolans and electric furnace slags), combined with 20% recycled concrete aggregates, still generally present improved mechanical properties. Regarding durability, depending on the exposure conditions related to corrosion risk induced by chloride ions or by carbonation, these concrete mixtures can lead to significant improvements in terms of chloride resistance and to a minor influence, or a slight decrease, in carbonation resistance, even though they have a generally long service life.

## Figures and Tables

**Figure 1 materials-14-04173-f001:**
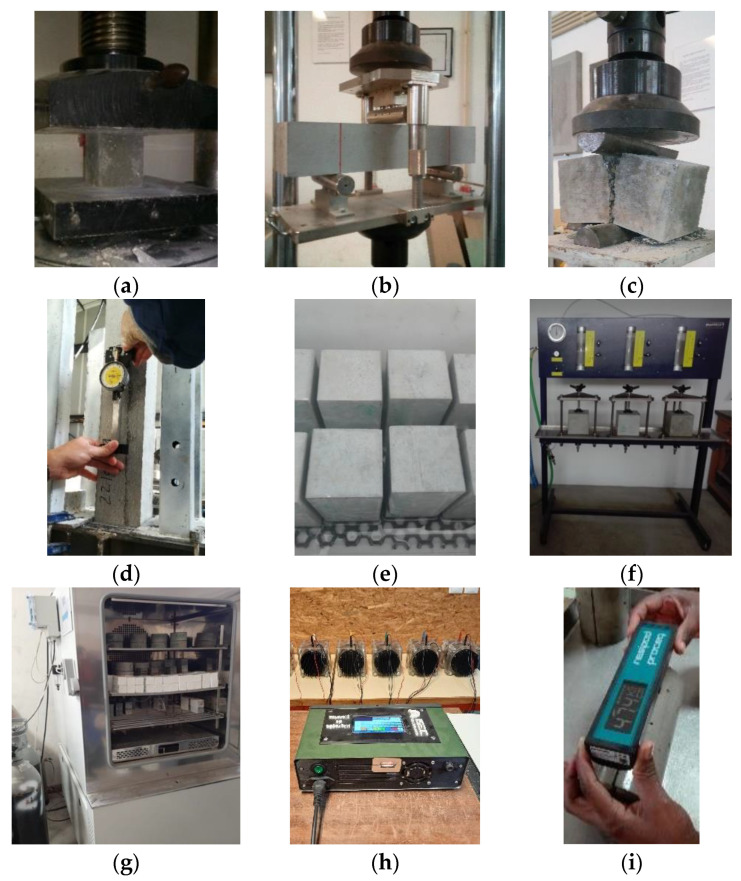
Tests to characterize the mechanical properties and durability: (**a**) compressive strength; (**b**) flexural strength; (**c**) tensile splitting strength; (**d**) Young’s modulus; (**e**) capillary absorption; (**f**) water penetration under pressure; (**g**) carbonation resistance; (**h**) chloride ion diffusion by non-steady state migration; (**i**) surface electrical resistivity.

**Figure 2 materials-14-04173-f002:**
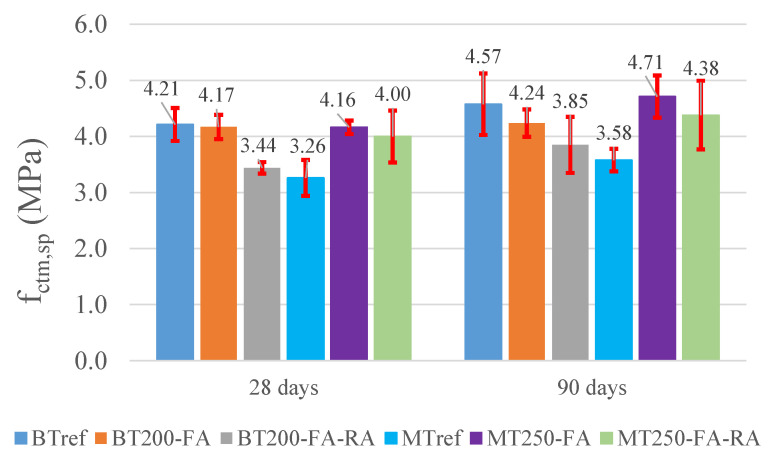
Tensile splitting strength for BT and MT mixtures, at 28 and 90 days.

**Figure 3 materials-14-04173-f003:**
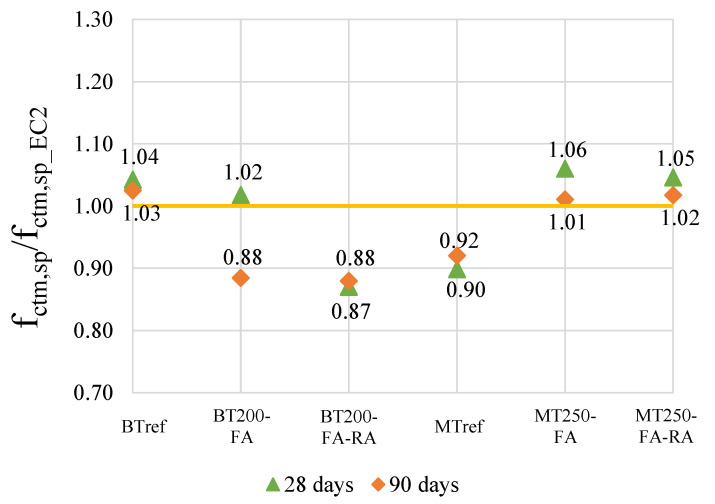
Ratio between tensile splitting strength tested values and those predicted in EC2.

**Figure 4 materials-14-04173-f004:**
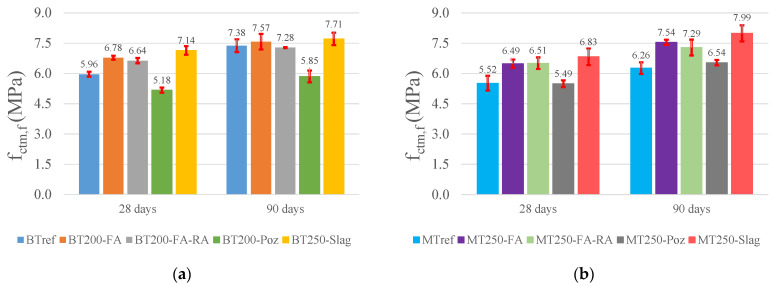
Flexural strength at 28 and 90 days: (**a**) BT mixtures; (**b**) MT mixtures.

**Figure 5 materials-14-04173-f005:**
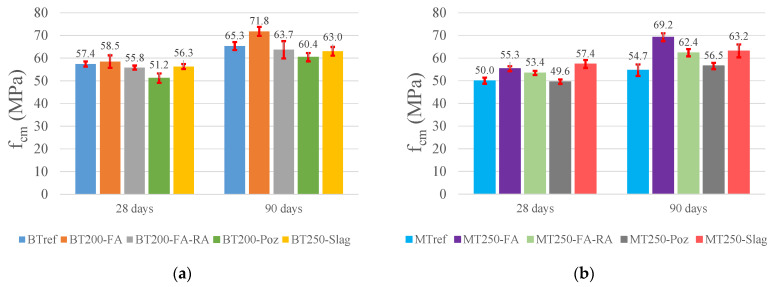
Compressive strength at 28 and 90 days: (**a**) BT mixtures; (**b**) MT mixtures.

**Figure 6 materials-14-04173-f006:**
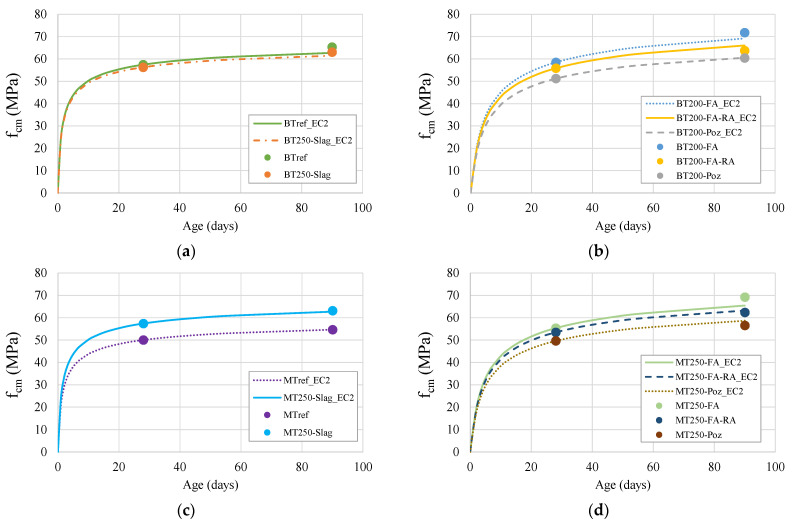
Evolution of average compressive strength with age and the influence of hardening in: (**a**) BT mixtures without pozzolanic addition; (**b**) BT mixtures with pozzolanic additions; (**c**) MT mixtures without pozzolanic additions; (**d**) MT mixtures with pozzolanic additions.

**Figure 7 materials-14-04173-f007:**
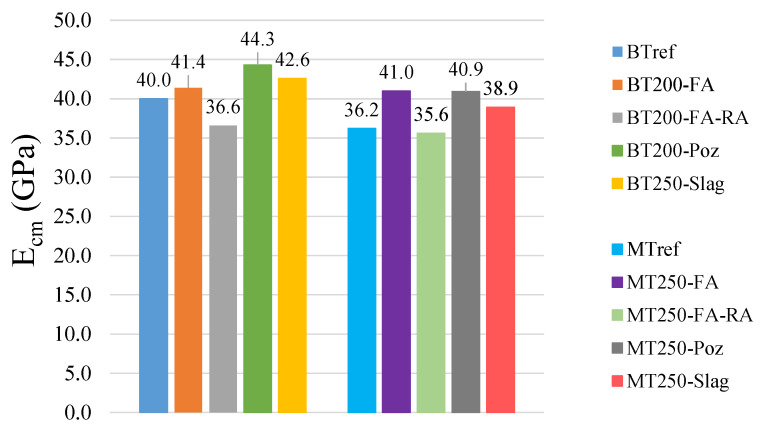
Young’s modulus results for BT and MT mixtures, at 28 days.

**Figure 8 materials-14-04173-f008:**
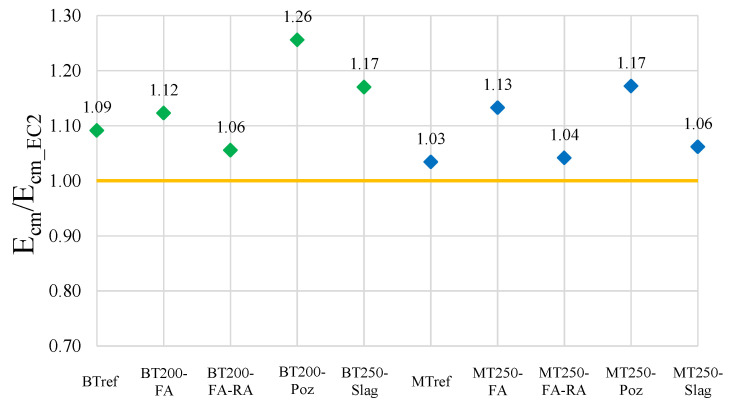
Ratio between Young’s modulus tested values and those predicted in EC2.

**Figure 9 materials-14-04173-f009:**
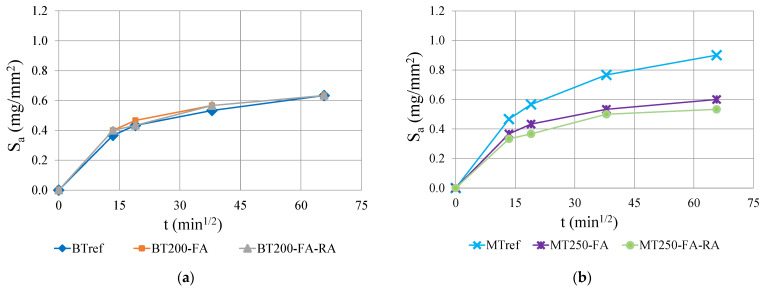
Capillarity water absorption through square root of time: (**a**) BT mixtures; (**b**) MT mixtures.

**Figure 10 materials-14-04173-f010:**
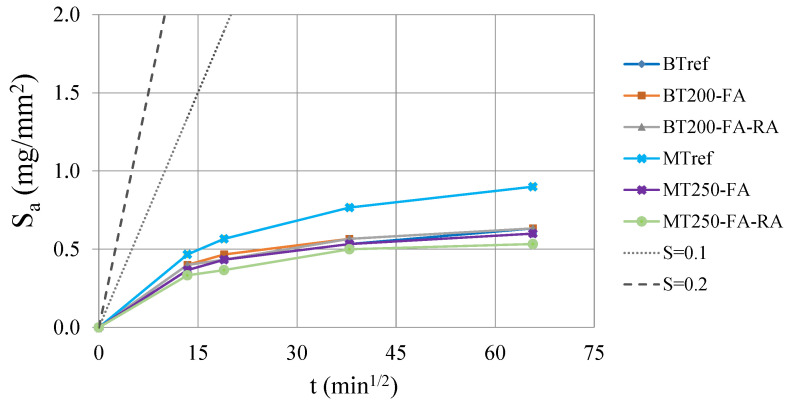
Capillary water absorption through square root of time.

**Figure 11 materials-14-04173-f011:**
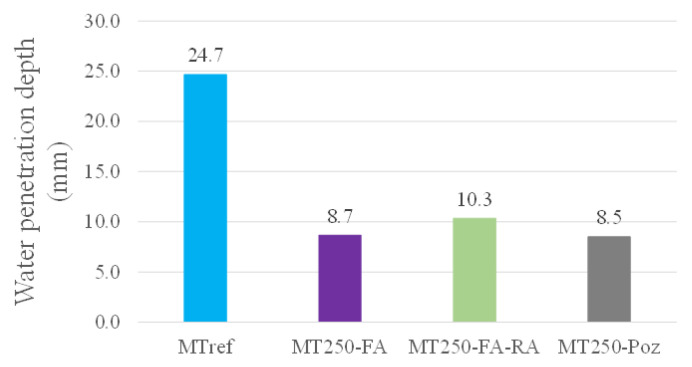
Water penetration depth under pressure for MT mixtures, at 28 days.

**Figure 12 materials-14-04173-f012:**
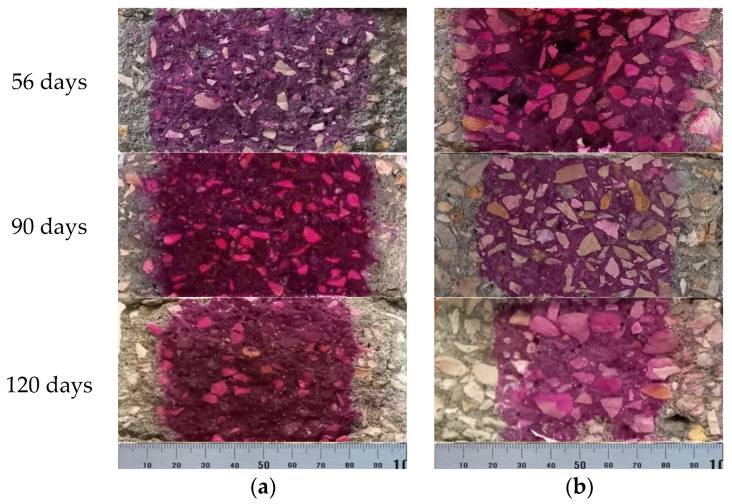
Specimens tested regarding resistance to carbonation over exposure time: (**a**) BT200-FA-RA; (**b**) MT250-slag.

**Figure 13 materials-14-04173-f013:**
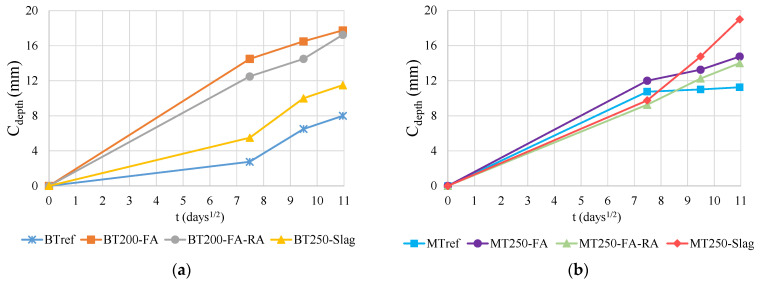
Carbonation depth through square root of time: (**a**) BT mixtures; (**b**) MT mixtures.

**Figure 14 materials-14-04173-f014:**
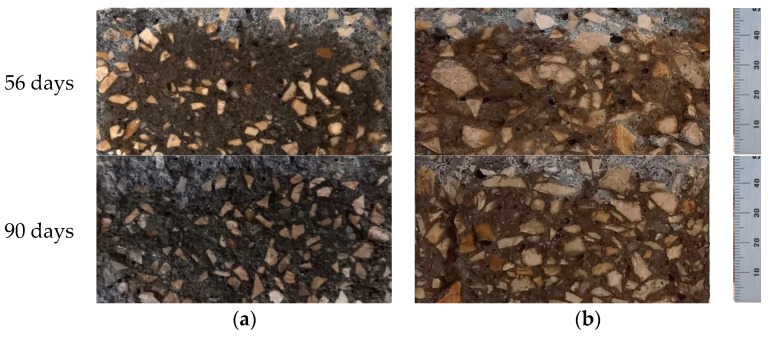
Specimens tested for resistance to chloride migration, at 56 and 90 days: (**a**) BT200-FA; (**b**) MT250-Poz.

**Figure 15 materials-14-04173-f015:**
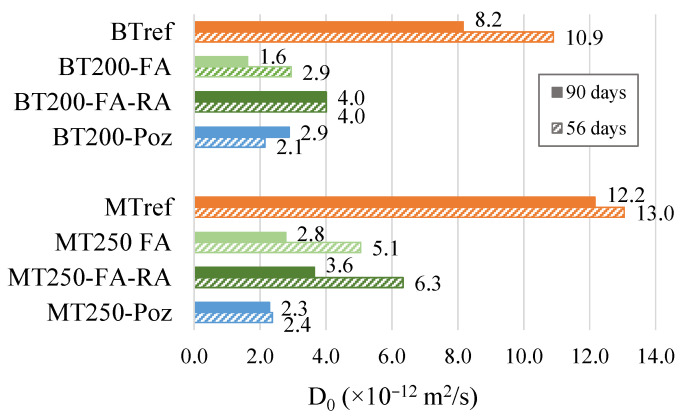
Chloride diffusion coefficient (D_0_), at 56 and 90 days, for BT and MT mixtures.

**Figure 16 materials-14-04173-f016:**
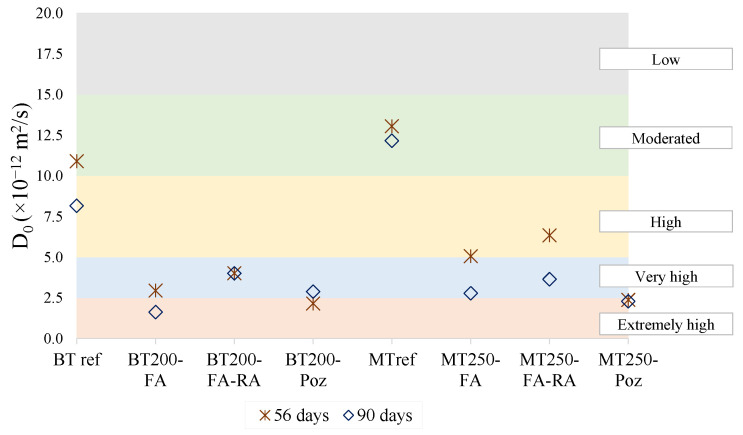
Chloride penetration resistance of concrete depending on the chloride diffusion coefficient (D_0_).

**Figure 17 materials-14-04173-f017:**
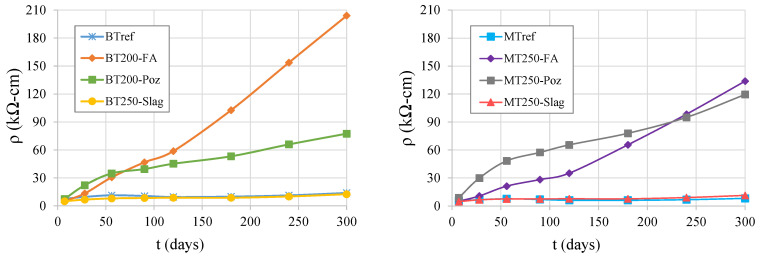
Concrete surface resistivity over time: BT mixtures (**left**); MT mixtures (**right**).

**Figure 18 materials-14-04173-f018:**
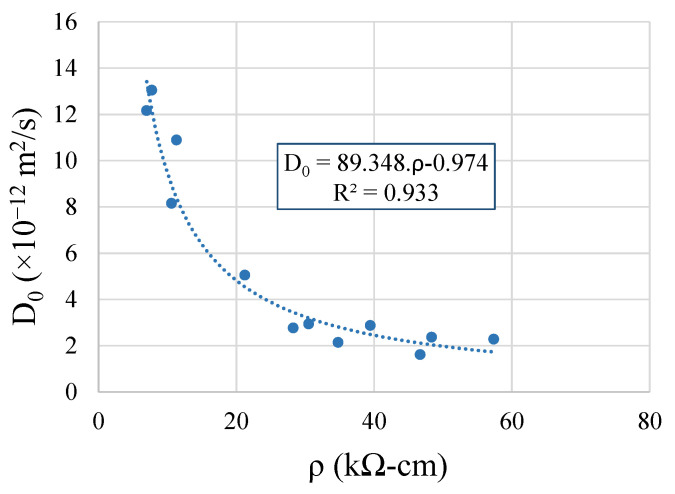
Correlation between the chloride diffusion coefficient and the surface resistivity.

**Figure 19 materials-14-04173-f019:**
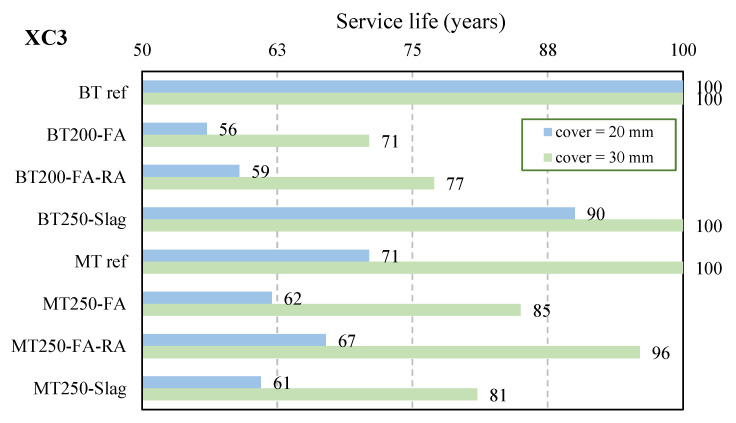
Service life expected for poles, class XC3.

**Figure 20 materials-14-04173-f020:**
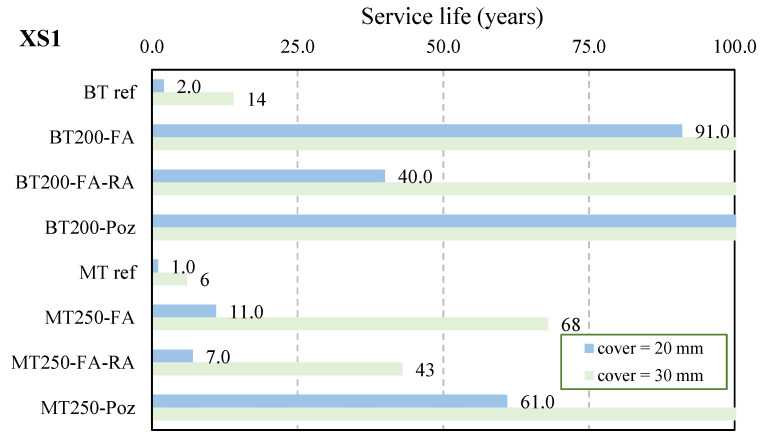
Service life expected for poles, class XS1.

**Table 1 materials-14-04173-t001:** Chemical composition of the powders, characterized by XRF (X-ray fluorescence).

Test	C	FA	Poz	Slag
% SiO_2_	17.38	54.0	46.18	16.10
% Al2 O_3_	4.67	22.0	16.44	11.08
% Fe_2_O_3_	2.91	8.50	6.30	31.75
% CaO	62.00	6.00	6.11	27.31
% MgO	1.38	1.60	3.09	5.71
% SO_3_	2.96	0.00	0.01	0.31
% K2O	0.54	1.60	4.19	0.02
% Na_2_O	0.09	1.00	5.80	0.07
% TiO_2_	-	1.20	1.91	0.67
% P_2_O_5_	-	0.80	0.41	0.50
% MnO	-	<0.3 (l.q)	0.26	3.53
% SrO	-	-	0.08	0.03
% CuO	-	-	-	0.04
% V_2_O_5_	-	-	-	0.11
% Cr_2_O_3_	-	-	-	1.92
% ZrO_2_	-	-	-	0.03
% Nb_2_O_5_	-	-	-	0.03
% BaO	-	-	-	0.12
% Cl	0.05	-	-	0.04
% F	-	-	-	0.65
LOI (%)	8.00	3.90	8.33	−2.08

**Table 2 materials-14-04173-t002:** Constituents proportioning and formulation parameters of the studied concrete mixtures.

Constituents	BTref	BT200-FA	BT200-FA-RA	BT200-Poz	BT250-Slag	MTref	MT250-FA	MT250-FA-RA	MT250-Poz	MT250-Slag
C (kg/m^3^)	400	200	200	200	250	360	250	250	250	250
FA (kg/m^3^)	-	150	150	-	-	-	100	100	-	-
Poz (kg/m^3^)	-	-	-	150	-	-	-	-	100	-
Slag (kg/m^3^)	-	-	-	-	100	-	-	-	-	100
HE (kg/m^3^)	-	7.0	9.1	7.5	7.5	-	-	-	-	-
Sky (kg/m^3^)	-	-	-	-	-	2.2	1.8	2.5	2.5	2.3
W_eff_ ^1^ (kg/m^3^)	160	109	112	114	113	173	143	143	143	143
W_abs_ ^2^ (kg/m^3^)	-	-	13	-	-	-	-	13	-	-
S0/1 (kg/m^3^)	-	402	230	398	430	-	270	221	270	286
S0/4 (kg/m^3^)	923	403	941	399	431	773	632	579	632	671
C2/5 (kg/m^3^)	898	1144	368	1138	1120	586	197	440	197	195
C5/12 (kg/m^3^)	-	-	-	-	-	457	795	272	795	787
RA1/8 (kg/m^3^)	-	-	336	-	-	-	-	-	-	-
RA4/16 (kg/m^3^)	-	-	-	-	-	-	-	329	-	-
Aggregates (kg/m^3^)	1822	1949	1873	1935	1981	1816	1894	1841	1894	1939
W/C	0.40	0.55	0.56	0.57	0.45	0.48	0.57	0.57	0.57	0.57
W/B	0.40	0.31	0.32	0.33	0.32	0.48	0.41	0.41	0.41	0.41
Compactness	0.820	0.865	0.860	0.860	0.860	0.805	0.840	0.840	0.840	0.840

^1^ Effective water; ^2^ absorption water.

**Table 3 materials-14-04173-t003:** Average values of carbonation resistance, R_C65_.

Concrete	R_C65_ (kg·year/m^5^)
BTref	1875.6
BT200-FA	160.7
BT200-FA-RA	195.6
BT250-Slag	601.4
MTref	357.8
MT250-FA	238.9
MT250-FA-RA	306.8
MT250-Slag	219.5

**Table 4 materials-14-04173-t004:** Values of t_p_ and t_ic_ for each exposure class, XC.

Periods during Service Life	XC2	XC3	XC4 (Dry Reg.)	XC4 (Wet Reg.)
t_p_ (years)	10	45	15	5
t_ic_ (years)	80	10	70	90

**Table 5 materials-14-04173-t005:** Minimum cover, c_min,dur_, to resist corrosion induced by carbonation in electrical concrete poles.

Concrete	Minimum Cover, c_min,dur_ (mm)			
	XC2	XC3	XC4 (Dry Reg.)	XC4 (Wet Reg.)
BTref	3	4	5	6
BT200-FA	9	14	19	21
BT200-FA-RA	8	12	17	19
BT250-Slag	5	7	10	11
MTref	6	9	12	14
MT250-FA	8	11	15	17
MT250-FA-RA	8	12	16	18
MT250-Slag	80	10	70	90

**Table 6 materials-14-04173-t006:** Prediction of the service life, tg, for different cover values (years), classes XC.

Concrete	Cover (mm)	XC2	XC3	XC4 (Dry Reg.)	XC4 (Wet Reg.)
BTref	15	>>100			
	20				
	30				
BT200-FA	15	>>100	51	35	25
	20		56	56	46
	30		71	>>100	
BT200-FA-RA	15	>>100	53	41	31
	20		59	67	57
	30		77	>>100	
BT250-Slag	15	>>100	69	117	107
	20		90	>>100	
	30		>>100		
MTref	15	>>100	59	69	59
	20		71	>>100	
	30	>>100			
MT250-FA	15	>>100	54	48	38
	20		62	82	72
	30		85	>>100	
MT250-FA-RA	15	>>100	57	60	50
	20		67	>>100	96
	30		96	>>100	
MT250-Slag	15	>>100	54	45	35
	20		61	75	65
	30		81	>>100	

**Table 7 materials-14-04173-t007:** Minimum cover, c_min,dur_, to resist corrosion induced by chlorides in electrical concrete poles.

Concrete	Minimum Cover, c_min,dur_ (mm)			
	XS1 ^1^	XS2 ^2^		XS3 ^3^
		1 m	1.4–25 m	
BTref	40	45	50	97
BT200-FA	17	21	24	46
BT200-FA-RA	21	25	29	55
BT200-Poz	15	18	21	40
MTref	48	52	58	112
MT250-FA	28	31	35	67
MT250-FA-RA	31	35	39	74
MT250-Poz	19	21	24	46

^1^ Poles exposed to the air sea salts located on the coast; ^2^ permanently submerged; ^3^ poles in tidal splash and spray zones.

**Table 8 materials-14-04173-t008:** Prediction of the service life, t_g_, for different cover values (years), classes XS.

Concrete	Cover (mm)	XS1	XS2		XS3
			1 m	1.4–25 m	
BTref	20	2	41	40	0
	30	14	42	42	0
BT200-FA	20	91	48	45	1
	30	>>100	76	63	7
BT200-FA-RA	20	40	44	43	1
	30	>>100	59	52	3
BT250-Poz	20	>>100	54	49	2
	30	>>100	101	79	14
MTref	20	1	40	40	0
	30	6	41	41	0
MT250-FA	20	11	42	41	0
	30	68	49	46	1
MT250-FA-RA	20	7	41	41	0
	30	43	46	44	1
MT250-Poz	20	61	48	45	1
	30	>>100	75	64	8

## Data Availability

Data available on request due to restrictions eg privacy or ethical. The data presented in this study are available on request from the corresponding author. The data are not publicly available due to confidentiality restrictions of the funded project.

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
