# Peer review of "Influence of Pozzolan, Slag and Recycled Aggregates on the Mechanical and Durability Properties of Low Cement Concrete"

_materials, 2021, doi:10.3390/ma14154173_

Round 1

Reviewer 1 Report

  1. The current study investigates the applicability of improving the performance of concrete that is made from materials to replace the usage of cement by using other materials such as fly ash, natural pozzolans and electric furnace slags. For this, the authors study several mechanical properties. The authors report improvement in mechanical properties and durability than traditional Portland cement-based mixtures.
  2. The abstract reads well but needs further improvement, the most important is that the main findings are not clearly stated, the ones mentioned now are simple and do not provide useful information on which material mixtures provided the best properties and by how much compared to other mixtures. Therefore, please consider reviewing the abstract and highlight the novelty, major findings and conclusions.
  3. The introduction reads well, however, the authors are encouraged to answer the following question after line 135.
  4. Also, after line 135, the authors must again briefly summarise what was done in the current work (this is a standard in all scientific articles).
  5. The authors are encouraged to combine figures 1 and 2 into one larger figure.
  6. Line 245 “the flexural strength was not affected by the incorporation of RA” why, can the authors elaborate further on this finding and try to support it by referring to previous studies in the open literature.
  7. In figures 3 and 5 can the authors unify the colours, so they are same for each of the mixtures in all the figures. (Keep consistency).
  8. Combine lines 274-275 with the previous paragraph.
  9. Line 436-438 can you support this claim by references from the open literature or known knowledge from the field of cement and construction in any books..etc.
  10. Line 484-485 again please explain this conclusion/claim, why the resistivity increases? And support with references if possible (i.e., what did previous studies find, was it similar to your results or different, in any case please discuss further).
  11. The authors should add a list of nomenclature for all the symbols and Greek letters reported in this work at the end of the manuscript.
  12. Overall, good study and can be accepted for publication.

Reviewer 2 Report

Half of a paragraph (lines 106-135) should be moved to the chapter “Materials and methods”.

What does FRX stand for in the first table?

There is a need for an explanation how 1 Effective water; 2 Absorption water are determined in the second table.

The second part does not describe how the mixture was mixed and how the samples were thickened.

Lines 213-223 lack an explanation for why properties were determined for only 6 sample series and not all 10.

Line 228: Do EC2[65] and MC10 differ from one another in any way? Why are both written but Figure 4 shows only EC2? Then what differences do they have?

Why was Capillarity water absorption not determined for all samples?

Why was Water penetration depth under pressure not determined for samples MT250-Poz when talking about the pozzolanic effect?

Lines 508-514 should be moved to “Methods and materials”

Reviewer 3 Report

The article is very interesting and concerns the influence of pozzolan, slag, and recycled aggregates on the mechanical and strength properties of low-cement concrete. Numerous tests of mechanical properties (compressive strength, bending strength, tensile strength, and Young modulus) and durability tests (capillary absorption, water penetration under pressure, carbonation resistance, chloride ion diffusion by non-steady-state migration, surface electrical resistivity) were performed.

The reviewer has the following comments:

  1. Please describe the differences in more detail and indicate the reason for the differences between the results of the tests of mechanical properties after 28 and 90 days from concreting the samples. The influence varies depending on the additives used. Maybe it would be good to make an efficiency map?
  2. The reviewer has some doubts about the carbonation tests. First of all, it should be noted that the phenolphthalein test is a rough estimate. Much better results would be obtained by pulverizing the concrete cover and performing the pH meter test. In addition, there are various electrochemical methods for carbonation testing. However, obtaining carbonation at a depth of several millimetres after 120 days suggests the use of an accelerated method. In fig. 2c you can probably see the carbonation chamber. The reviewer, therefore, asks for a more detailed description of the carbonation test process.
  3. While the standards from the Eurocode package are widely available, national standards are difficult to obtain and are usually written in languages ​​other than English. The reviewer is unfamiliar with the standards LNEC E-391, LNEC E-393, LNEC E-463, LNEC E-465, as are probably the majority of future readers of the article. Therefore, I am asking for the description of test procedures performed according to these standards to be developed.
  4. Is the construction of the hardening curve based on 2 points (Fig. 7), not too big an approximation?

Round 2

Reviewer 1 Report

All questions answered and paper can be accepted.